# Pharmacologic Rate versus Rhythm Control for Atrial Fibrillation in Heart Failure Patients

**DOI:** 10.3390/medicina58060743

**Published:** 2022-05-30

**Authors:** Ioanna Koniari, Eleni Artopoulou, Dimitrios Velissaris, Virginia Mplani, Maria Anastasopoulou, Nicholas Kounis, Cesare de Gregorio, Grigorios Tsigkas, Arun Karunakaran, Panagiotis Plotas, Ignatios Ikonomidis

**Affiliations:** 1Department of Cardiology, University Hospital of South Manchester NHS Foundation Trust, Manchester M23 9LT, UK; iokoniari@yahoo.gr (I.K.); arun.karunakaran2@nhs.net (A.K.); 2Department of Internal Medicine, University Hospital of Patras, 26504 Patras, Greece; helenartopoulou@gmail.com (E.A.); dimitrisvelissaris@yahoo.com (D.V.); 3Department of Cardiology, University Hospital of Patras, 26504 Patras, Greece; virginiamplani@yahoo.gr (V.M.); anastmaria89@hotmail.com (M.A.); ngkounis@otenet.gr (N.K.); grigoriostsigkas@gmail.com (G.T.); 4Department of Clinical and Experimental Medicine Cardiology Unit, University Hospital of Messina, 98125 Messina, Italy; degregorio@unime.it; 5Laboratory Primary Health Care, School of Health Rehabilitation Sciences, University of Patras, 26504 Patras, Greece; pplotas@upatras.gr; 6Second Cardiology Department, Attikon University Hospital, Medical School, National and Kapodistrian University of Athens, 12462 Athens, Greece

**Keywords:** atrialfibrillation, heart failure, rate control, rhythm control, treatment strategy, studies

## Abstract

Atrial fibrillation (AF) and Heart failure (HF) constitute two frequently coexisting cardiovascular diseases, with a great volume of the scientific research referring to strategies and guidelines associated with the best management of patients suffering from either of the two or both of these entities. The common pathophysiological paths, the adverse outcomes, the hospitalization rates, and the mortality rates that occur from various reports and trials indicate that a targeted therapy to the common background of these cardiovascular conditions may reverse the progression of their interrelating development. Among other optimal treatments concerning the prevalence of both AF and HF, the introduction of rhythm and rate control strategies in the guidelines has underlined the importance of sinus rhythm and heart rate control in the prevention of deleterious complications. The use of these strategies in the clinical practice has led to a debate about the superiority of rhythm versus rate control. The current guidelines as well as the published randomized trials and studies have not proved that rhythm control is more beneficial than the rate control treatments in the terms of survival, all-cause mortality, hospitalization rates, and quality of life. Therefore, the current therapeutic strategy is based on the therapy guidelines and the clinical judgment and experience. The aim of this review was to elucidate the endpoints of pharmacologic randomized clinical trials and the clinical data of each antiarrhythmic or rate-limiting medication, so as to promote their effective, individualized, evidence-based clinical use.

## 1. Introduction

Atrial fibrillation (AF) and heart failure (HF) are evolving as new cardiovascular epidemics over the last decade [1]. AF is the most common atrial arrhythmia in clinical practice and the most common sustained arrhythmia seen in HF population [1]. Moreover, the incidence of HF has been highly increased over the past few years [2].

Research has been done in the aspects of HF and AF, although the approaches that will offer the best clinical outcomes still remain unclear. The suggested common pathophysiological pathways that these two entities share, introduce combined therapeutic strategies in order to eliminate the prevalence of adverse outcomes resulting from their concomitant existence. Rate and rhythm control medication are considered to be primary choices in the treatment of patients with coexisting HF and AF [2]. The effective application of these treatments and the superiority of either strategy over the other remain the subject of many trials and hold the clinical practitioners’ interest [1].

The aim of this comprehensive review was to demonstrate the interdependent pathophysiological mechanisms of HF and AF and further analyze the landmark clinical trials of current rate and rhythm control pharmacotherapies, reflecting their challenges in daily clinical practice. In addition, we discuss extensively the pharmacologic randomized clinical trials, elucidating the endpoints, strengths, and potential limitations of either strategy and also the clinical data of each antiarrhythmic or rate-limiting medication separately, so as to promote their effective, individualized evidence-based clinical use. Recent trials of catheter ablation are promising in this subgroup of patients; however, the scope of this review was to focus on pharmacologic guideline-directed rhythm and rate-control strategies.

## 2. Interrelated Pathophysiology of HF and AF 

The pathophysiological relationship between these two entities is not entirely clear and still remains a matter of research, but their coexistence in a great percentage of the population is beyond doubt [1]. This can be explained to some extent due to the common risk factors such as diabetes, ischemic and nonischemic heart disease, hypertension, and growing age [2]. There is a plethora of mechanisms by which HF can lead to AF, such as atrial pressure overload, atrial enlargement, structural remodeling, and oxidative stress. The elevation of left atrial pressure either chronically or acutely, can lead to atrial fibrosis and distension, resulting in conduction abnormalities [3]. On the other hand, AF can be the cause for the HF development [1,2]. Rapid ventricular rates as well as a lack of atrial systole result in elevated atrial pressures and decreased cardiac output [1,2,4]. HF exacerbations can be attributed to the elevation of left atrial pressure. Coronary artery disease, arterial hypertension, diabetes, obesity, and increasing age are considered to be common factors causing chronic inflammation and leading both to atrial fibrillation as well as heart failure [5]. Moreover, AF can also cause the development of HF, through tachycardia-mediated cardiomyopathy [2]. In this case, ventricular dysfunction is caused by persistent rapid ventricular rates, and it could be the result of any persistent atrial tachycardia, with AF being the most common one [6]. However, the diagnosis of this entity is a difficult one and remains a diagnosis of exclusion.

AF may coexist not only in HF with a reduced ejection fraction (HFrEF), but also in HF with preserved ejection fraction (HFpEF), with the same risk factors contributing to this coexistence. Interestingly, research has focused on AF occurring on HFrEF, basically including persistent and long-standing AF, according to the population studied [2,7].

## 3. Management of AF in HF Patients

Managing AF in the HF patients is linked with several therapeutic challenges. There are several contraindications to medications that are usually administrated in AF and the patients’ response to the used treatment for AF may not be the desired one. Additionally, the association between AF and acute HF may not be so easily clarified, whereas the therapeutic goals in patients where the two conditions are coexisting are not clear yet [8]. The major complication in patients with atrial fibrillation is an ischemic stroke. The targets of treatment are to prevent stroke, sustain sinus rhythm as long as possible, control ventricular rate in long-standing and permanent atrial fibrillation, and, at last, to eliminate symptoms and improve the quality of life (QoL). The cornerstone of the treatment is considered to be anticoagulation. Guidelines recommend initiation of anticoagulation based on the CHA_2_ DS_2_VaSc risk assessment score. All subtypes of AF must be treated with anticoagulants if CHA_2_ DS_2_VaSc is greater than 2 points in men and three points in women, and they should be treated with anticoagulants if CHA_2_ DS_2_VaSc is one point in men and two points in women. The selection of the type of anticoagulant is based in patients’ clinical characteristics and personal wishes.

Clinical characteristics are also the main factor for the clinician’s decision to treat AF with rate or rhythm control. Multiple factors can affect this decision such as patient’s age, duration of AF, echocardiographic characteristics of the left atrium, medical comorbidities, and of course the presence of symptoms. Rhythm control can be achieved using Class I or Class III antiarrhythmics. Class Ic medicines should be avoided in patients with ventricular dysfunction, or patients with a history of coronary disease due to their proarrhythmic effect. On the other hand, Class III antiarrhythmics can be used safely in these patients but with the risk of non-cardiac complications. If one chooses rate control, then beta blockade is the cornerstone of this therapy. 

Several studies and sub-analysis have been conducted in order to determine the best case scenario between pharmacologic rate versus rhythm control on AF in HF patients. Yet, there is not a clear answer. 

Rate control is not without challenges [2]. There are no strict guidelines for the optimal heart rate and is up to the clinician whether following a strategy with a more lenient or not heart rate, based on patients’ symptoms and NYHA class. Heart rate is achieved using beta blockers, glycosides, and non-dihydropyridine calcium channel blockers, while beta blockers and glycosides are also used for the heart failure therapy [2]. From a different point of view, the long term use of antiarrhythmic drugs may contribute to the all-cause mortality, while maintaining sinus rhythm. Patients takingClass I medication are prone to its proarrhythmic effects, while Class III antiarrhythmics drugs are responsible for thyroid, respiratory, hepatic, and renal toxicity.

## 4. Rate vs. Rhythm Control 

Rapid ventricular rate, irregular ventricular response, and loss of atrial contraction are all associated with adverse hemodynamic events and a further worse prognosis in patients with AF [9,10,11,12,13,14,15,16,17,18,19,20]. The restoration of sinus rhythm is linked with improved cardiac output, exercise capacity, and maximal oxygen consumption [10,16]. According to the data occurring from prospective studies, the onset of AF in patients with CHF is associated with clinical and hemodynamic decline [19,20] and an increased mortality rate due to the embolic complications of AF [20,21,22,23,24], as well as serious complications associated with the use of antiarrhythmic drugs [25,26,27].

In the management of a patient with AF and HF, the administration of an initial treatment strategy with a subsequent alternative planning is essential. A rhythm control strategy targets the restoration and maintenance of sinus rhythm (SR) and the control of the ventricular response rate in the case of recurrent AF. A ratecontrol strategy targets mainly the control of the ventricular response rate [8]. While recent studies have shown that rate and rhythm control strategies in patients with AF and HF lead to similar outcomes, it is still unclear why certain patients may benefit from one strategy over the other [28].

In the following sub-sections, we present in detail the landmark randomized clinical trials comparing pharmacologic rhythm versus rate strategies, referring to enrolled patient characteristics, pharmaceutical agents used to achieve rhythm/rate control and/or potential adverse effects, primary and secondary endpoints regarding cardiovascular morbidity, mortality, QoL, HF hospitalizations/symptomatic improvement, thromboembolic, and bleeding complications.

### 4.1. Rate Control versus Electrical Cardioversion (RACE) Study

The RACE study included 522 patients with persistent AF observed from 1 June 1998, until 1 July 2001.In a sub-study on AF patients with mild to moderate chronic heart failure (CHF) with NYHA functional Classes II and III, 261 individuals were analyzed, while 130 and 131 patients were randomized to rate and rhythm control groups, respectively [29]. In this study, rate control was achieved using negative chronotropic drugs (digitalis, b-blockers, and nondihydropyridine calcium channel blockers). Rhythm control was achieved through serial electrical cardioversion along with the use of anticoagulation and the use of antiarrhythmic drugs (sotalol, Class Ic drugs, or amiodarone). Class Ic drugs were administrated in cases of mild CHF without the presence of coronary artery disease and current or preexisting CHF. The primary end point of the RACE study was the composite of cardiovascular death, hospitalization for CHF, thromboembolic complications, bleeding, pacemaker implantation, or severe adverse effects of antiarrhythmic drugs, as it occurred from a maximum of 3 years of follow up [30]. An assessment of quality of life (QoL) according to the Medical Outcomes Study Short-Form Health Survey (SF-36) questionnaire was included, based on the descriptions of the patients that completed the questionnaires at baseline and at the end of the follow-up [31,32] and, additionally, QoL was compared between the rate and rhythm control groups [29]. The objective of the RACE study was to show that there is no superiority of the two strategies (rate and rhythm control) according to the occurrence of the primary end point [30].

The results in the group of patients with AF and mild CHF showed that the occurrence of cardiovascular morbidity and mortality and QoL is equal between those treated with rate control and those treated with rhythm control. Between the two groups, however, there were important differences in the end points. In the rate control group, a higher mortality and morbidity, thromboembolic complications, and major fatal and nonfatal bleedings were observed. In the rhythm control group, a successful maintenance of sinus rhythm was associated with excellent survival whereas there was no improvement in the progression of CHF [29]. In another RACE substudy, whereas routine rate control prevented a decline in left ventricular function, the maintenance of sinus rhythm led to an improvement of left ventricular function and an attenuation of atrial size [33].

The administration of prophylactic antiarrhythmic drugs had significant effects on the end points in the rhythm control arm and not in the rate control arm. Although, one patient in sotalol treatment died suddenly, there were no reported severe adverse effects during this treatment. During treatment with Class Ic drugs, life-threatening adverse events were reported in four patients, in one patient during the institution of the drug in the hospital and additionally two patients died suddenly while being in Class Ic drugs treatment. Cardiovascular mortality in patients not treated with Class Ic antiarrhythmics was reported equally among the whole study population, with more events reported in the rate control group [29]. Amiodarone was generally safe in administration and only caused nonfatal bradyarrhythmias [29], as it was not associated with a negative impact on survival in patients with CHF [34]. In patients with AF and CHF, it has been shown by RACE study and other previous observations that amiodarone is an effective and safe antiarrhythmic drug [34,35,36], which is associated with a lower mortality rate in AF patients at baseline where SR was restored [35]. Additionally, no differences were observed in QoL at baseline between rate and rhythm control groups, during long-term treatment and at the end of follow-up. Overall, SR maintenance was associated with QoL improvement, as well as a development of exercise tolerance and sense of vitality (Table 1) [29].

### 4.2. Atrial Fibrillation and Congestive Heart Failure (AF-CHF) Trial

The AF-CHF study included 90 cardiology centers throughout Canada, the United States, South America, and Europe. The recruitment started in May 2001, the randomization was concluded in June 2005, and the follow-up period ended on 30June 2007 [37].

The patients enrolled had a left ventricular ejection fraction of 35% or less, as well as a history of congestive heart failure and atrial fibrillation and a required long-term therapy in either of the two study groups where the patients were randomly assigned, either to the rhythm control group or the rate control group [37]. Rhythm control was achieved through the administration of antiarrhythmic drug therapy (amiodarone, sotalol, or dofetilide), the implantation of a permanent pacemaker in case of bradycardia, and electrical cardioversion in patients who did not have conversion to sinus rhythm after the administration of antiarrhythmic drugs [37]. Rate control was achieved through the administration of adjusted doses of beta-blockers with digitalis while atrioventricular nodal ablation and pacemaker therapy were applied on patients who did not respond to the ratecontrol therapy [37]. The primary outcome of the study was death from cardiovascular causes and secondary outcomes were death from any cause, stroke, worsening congestive HF, hospitalization, quality of life, cost of therapy, and a composite of death from cardiovascular causes, stroke, or worsening congestive HF [37]. The results from the trial showed that in patients with AF and congestive HF, the routine use of a rhythm control strategy did not reduce the rate of death from cardiovascular causes and there were no significant differences in important secondary outcomes (death from any cause, worsening heart failure, or stroke), as compared with a rate control strategy. In this trial, patients receiving Class I antiarrhythmic agents were excluded and there was a higher rate of warfarin use when compared to the previous trials. The results of this trial could not be extended in patients with HFpEF. Patients in the rhythm control group had a higher possibility of hospitalization than those in the rate control group especially during the first year after enrolment. Additionally, the potential benefit of SR maintenance in the mortality rate may be affected by the harmful effects of the administrated antiarrhythmic therapies. In conclusion, this clinical trial showed no superiority of a rhythm control strategy against a rate control one, while the latter eliminated the need for repeated cardioversion and reduced rates of hospitalization, which led to the suggestion that rate control treatment should be considered as a primary approach for patients with AF and congestive HF (Table 1) [37].

### 4.3. Atrial Fibrillation Follow-Up Investigation of Rhythm Management (AFFIRM) Trial

During the AFFIRM trial, 4060 patients with AF and HF were analyzed [38]. The enrollment in the study began on 9 November 1995, and was concluded on 31 October 1999, and the follow-up ended on 31 October 2001. The study aimed at the comparison of rate and rhythm control treatment strategies and the primary end point was the definition of the all-cause mortality rates. Anticoagulation therapy was administered to both patient groups [39]. Patients in the rate control arm of the study were treated with an AV nodal-blocking drug randomly chosen by the treating team: b-blockers, calcium channel blockers, or digoxin, alone or in combination [40]. Patients in the rhythm control arm of the study were treated with any of the available antiarrhythmic drugs, with the dosages carefully adjusted for renal and hepatic dysfunction. Class I drugs were carefully administered in patients with left ventricular dysfunction and in those with ischemic heart disease. Class Ic agents (flecainide and propafenone) were not used in patients with any evidence of left ventricular dysfunction, congestive heart failure, left ventricular hypertrophy, coronary artery disease, or myocardial ischemia or infarction. Sotalol and disopyramide were also not administered in patients with evidence of severe left ventricular dysfunction. Amiodarone was mainly used in the elderly population, but due to its adverse effects, it was not preferable in the treatment of younger patients [39].

In an intention-to-treat analysis by Guglin et al. [38], the patients were randomized in groups according to the treatment strategy: 1779 in the rate control group, 1439 in the rhythm control group, and 4 crossover groups; 162 patients that crossed over from rate to rhythm control, 533 patients that crossed over from rhythm to rate control, and 147 patients who crossed over twice in two different groups, from rate to rhythm and back to rate, and from rhythm to rate and back to rhythm. In each group, functional status by NYHA (0, I, II, III; there were no Class IV patients) was estimated at baseline and at each follow-up visit as well as the rhythm at the time of the visit. For analysis purposes, the patients belonging to the NYHA functional Classes 0 and I were combined in a group of asymptomatic patients, and those in NYHA Classes II and III were combined as symptomatic HF patients. In the original AFFIRM results presentation, it was reported that uncontrolled symptoms due to AF and HF were the most common reasons for the initial crossover from rate to rhythm control, and an inability to maintain SR and drug intolerance were the most common reasons for the crossover from rhythm to the rate control strategy [41]. Guglin et al. [38] reevaluated the original analysis, reporting that the crossover from rate to rhythm control groups was observed mainly in heavily symptomatic AF patients (64.8%) or in patients with new-onset or worsened HF or with the development of proarrhythmic or other effects (29.6%), something that was also involved in the main reasons for the crossover from the rhythm to the rate control group (29.1%) along with the failure in achievement of SR in the rhythm control arm of the study (49.2%), as well as the development of intolerable adverse effects (19.5%). 

Patients in the rate control arm were more frequently in AF, while patients in the rhythm control arm were in SR about 11 times more frequently than in AF. The two groups with the lowest frequency of SR were associated with much worse functional status: rate control and rhythm-to-rate crossovers [38]. Patients who crossed over from rate to rhythm control had more eligible episodes of AF lasting fewer than 2 days, as it has previously been described by Curtis et al. [42] Patients in AF had a higher NYHA class than those in SR, those who switched from the rhythm to the rate control strategy presented with no significant differences in NYHA functional status, and those who switched strategy more than once had much worse symptoms both in SR and in AF when compared with other groups. Patients of all groups had fewer HF symptoms and required less HF medication (ACE inhibitors and diuretics) when they were in SR when compared with AF, except for those who crossed over from the rhythm to the rate control strategy, where the maintenance of SR was not possible to achieve [38].

The AFFIRM trial indicated that there is no survival benefit from the rhythm control strategy when compared to the rate control strategy [38]. A subanalysis by Freudenberger et al. on patients with left ventricular systolic dysfunction in the AFFIRM trial came to the conclusion that regarding mortality, hospitalization, and NYHA class, neither of the two strategies is more beneficial that the other regardless of baseline, moderate, or severe left ventricular dysfunction [43]. In an on-treatment analysis of the AFFIRM study, the maintenance of SR was associated with a decrease in the mortality rate [44]. An analysis of the patients participating in the AFFIRM trial not by strategy but by the actual rhythm indicated that the functional status was significantly benefited by the presence of normal SR [45]. In that study, however, Chung et al., calculated the mean NYHA class and as a result there was no difference found in the NYHA functional class status between the two strategy arms, despite the fact that SR was basically maintained in the rhythm control arm [45]. Deterioration in NYHA functional class at the end of the study was observed in all groups except in those patients who crossed over from rate to rhythm control, completing a previous suggestion where no differences between the two groups were observed [38,45]. It has been suggested that AF does not cause the functional decline directly, but it indicates a poorer cardiac status [38]. Guglin et al. demonstrated for the first time that the treatment arm concerning the rate control strategy, included the patients with highest prevalence of AF who also presented with a higher NYHA class throughout all follow-up visits [38]. 

In conclusion, according to the AFFIRM study, symptomatic HF was more common in the rate control than in the rhythm control arm, without introducing however a more beneficial profile of either strategy. In general, AF is shown to be associated with poorer NYHA functional class, more symptomatic HF, as well as with greater requirements for ACE inhibitors and diuretics, in all subgroups of the AFFIRM trial except for those patients who crossed over from rhythm to rate control, while the highest rate of symptomatic HF patients was reported among patients who changed strategy between rate and rhythm control more than once. This group of patients may represent a primary target for other treatment strategies such as ablation because of the greater benefit for them being in stable SR [38]. Additionally, the AFFIRM study showed that SR maintenance was not beneficial in the prevention of thromboembolic events and that long-term warfarin therapy should be included in the treatment of patients who are under rhythm control if they have additional risk factors for stroke (Table 1) [46].

### 4.4. CAFÉ-II Study

This randomized controlled study aimed to determine whether restoration and maintenance of SR were beneficial for patients with HF and persistent AF. The study included patients aged >18 years of age with persistent AF and chronic symptomatic HF (NYHA > Class II symptoms) with echocardiographic evidence of systolic dysfunction. No patients presenting contraindications in oral anticoagulants were included in the study. Patients were randomized to either a rate control or rhythm control strategy. Between follow-up visits the patients’ symptoms were recorded and additionally their quality of life (QoL) was assessed by specific questionnaires [47]. Digoxin and b-blockers were used to achieve rate control [47,48]. Rhythm control was achieved through the oral administration of amiodarone therapy and, after the persistence of AF after 2 months of treatment, external biphasic electrical cardioversion under general anesthesia was performed. A change to the rate control strategy was preferred if the attempts for restoring the SR through the rhythm control strategy were not successful. All patients were anticoagulated using warfarin, aiming for an International Normalized Ratio (INR) of 2.0 to 3.0 throughout the study [47].

The results of the study showed that a rhythm control strategy can improve QoL and left ventricular function in comparison with rate control strategy alone. For the patients whose SR was maintained at 1 year follow-up, they presented with the greatest improvement [47]. HF patients are associated with a higher recurrence of AF recurrence because of the existence of structural heart disease and dilated atria, which stimulate the development of AF [49]. Considering that a rhythm control strategy for AF is linked with the possibility of arrhythmia recurrence, any attempts of cardioversion in HF patients should be accompanied with additional treatment strategy to reduce this possibility. Amiodarone is reported to be a preferable choice for patients with concomitant HF and AF, as it increases the effectiveness of cardioversion and reduces the recurrence of AF [35,50]. In the CAFÉ-II study, SR was restored and maintained at 1 year (80 and 66% respectively) in patients treated with cardioversion and amiodarone or with amiodarone alone [47], although it has been reported that amiodarone may worsen the HF status by negatively affecting the long term mortality rate in patients with moderate or severe HF and left ventricular systolic dysfunction [51,52]. This study proved that the restoration and maintenance of SR in patients with left ventricular impairment improved cardiac function by restoring the atrial contribution to ventricular filling, regulating the cardiac cycle and improving the resting heart rate and chronotropic response to exercise. Restored SR was also associated through this study with a decrease in the levels of natriuretic peptides [47]. Exercise performance and NYHA functional class were similar between groups at 1 year, but it was suggested that patients who maintained SR at 1 year might be presented with an improvement in the above characteristics [47]. The improvement in left ventricular function and QoL achieved with rhythm control were not linked with an improvement in exercise capacity. In contrast to the also-not-clear results presented by the application of 6minwalktest, formal cardiopulmonary exercise testing may provide a better evaluation of the effect of rhythm restoration on exercise capacity. The CAFÉ-II study offers no results in the effect of the intervention in the mortality rates, while it suggests an improvement of the HF symptoms through cardioversion, offering complementary data in the multicenter AF-CHF study, which reported that this strategy is also safe but does not improve long-term morbidity or mortality [47]. These combined data offer a better aspect for the administration of the best treatment strategy for each individual patient (Table 1).

### 4.5. How to Treat Chronic Atrial Fibrillation (HOT CAFE) Study

The How to Treat Chronic Atrial Fibrillation (HOT CAFE) study described the comparison between the use of either the strategy of SR restoration and maintenance and the strategy of ventricular rate control and chronic thromboembolic prophylaxis in patients with persistent AF [53]. The recruitment started in March 1997, randomization was completed in December 2000, and the follow-up in December 2002. Of the 738 screened patients, 205 patients (134 men and 71 women) were enrolled into the study. Among the patients studied, HF patients were included except for those with severe cardiac disability, declared as NYHA functional Class IV. The conduction of the trial was based on an intention-to-treat protocol [53]. The rate control strategy included oral thromboembolic prophylaxis (based on the thromboembolic risk profile, the used drugs were aspirin or ticlopidine, target INR = 2.5 (range, 2.0 to 3.0), which were recommended for patients at a high risk of stroke [54]) and rate control drugs: b-blockers, nondihydropyridine calcium blockers, digoxin, alone or in a combination, cardioversion, and atrioventricular junctional ablation with pacemaker placement were alternative non-pharmacologic strategies in resistant tachycardia. The rhythm control strategy included cardioversion prior to the antiarrhythmic drug therapy: propafenone, disopyramide, sotalol, amiodarone, Class I drugs on occasion, and additionally disopyramide or propafenone b-blockers in clinical indication [53]. A thromboembolic prophylaxis (acenocoumarol) was administered 4 weeks before cardioversion in this arm of the study to achieve an INR of 2.0 to 3.0 [53,54]. The primary end point of the study was a composite of all-cause mortality, thromboembolic complications, ischemic stroke, intracranial, or other major bleeding. The secondary end points included rate control and maintenance of SR, discontinuation of therapy, bleeding complications, hospitalization, development of congestive HF and exercise tolerance, and the association of echocardiographic parameters with the progress of the condition [53].

The results of the HOT CAFΕ study showed that the rate control of persistent AF is equivalent to rhythm control in terms of the primary endpoints of the study, supporting the results of other trials despite the rather younger population or the longer persistence of AF (for up to 2 years) [53].

The ventricular response control was in both groups easily accomplished, resulting in a good management of symptoms and exercise tolerance, emphasizing the importance of the ventricular rate restoration in AF patients. A better rate control and a significant decrease in mean heart rate during follow-up was achieved in the patients of the rhythm control group, while in the rate control group the ventricular response was satisfactory with no significant changes during follow-up. While at baseline there were no significant differences, patients in the rhythm control arm had a higher maximal workload and longer exercise tolerance than patients in the rate control arm at the end of follow-up [53]. The rhythm control strategy group was associated with an increase in left ventricular fractional shortening and a decrease in the dimensions of right and left atria during follow-up in the rhythm control group, while the opposite was found in the rate control group, suggesting another possible mechanism of SR maintenance. In contrast, the strategy of rate control was associated with significantly fewer hospitalizations and less new-onset arrhythmias. The risk of thromboembolism occurrence has not been proven to be eliminated by the restoration of SR, since there were reported cases of stroke in the SR restoration group, although in the rate control group thromboembolic prophylaxis was effective. Long-term anticoagulation therapy, however, is not a preferable strategy, despite the fact that the combination of anticoagulants with a rhythm control strategy after cardioversion might be associated with lower stroke rates, even in patients with a stable SR. The results of the HOT CAFE trial may useful in future meta-analyses along with the results of other studies (Table 1) [53].

### 4.6. Pharmacological Intervention in Atrial Fibrillation (PIAF) Trial

PIAF trial is an open, randomized pilot study aiming to compare two different treatment strategies in patients with persistent and symptomatic AF lasting between 7 days and 360 days [55]. The patients enrolled in the study were randomized in two different groups: GroupA patients were treated with a ventricular rate control drug primarily with diltiazem, and if this treatment failed the therapy was chosen by the treating physician, whereas in Group B, patients were treated with pharmacological methods and, if necessary, with electrical cardioversion followed by antiarrhythmic therapy, which included amiodarone and, in the case of AF recurrence, the therapy was decided by the treating physician [56]. All patients were receiving an anticoagulation therapy throughout the trial; the target INR range was 2.0–3.0. Of the 252 patients enrolled in the trial, 125 patients were randomized to Group A and 127 to Group B. Among the patients enrolled, HF patients were included, except for patients with congestive heart failure and NYHA functional Class IV. All patients were in follow-up for 12 months after randomization. Τhe primary endpoints of the study were the improvement of the AF outcomes and the reported symptoms. Other secondary study endpoints were the change in mean heart rate during AF, the stabilization of SR, the number of hospital admissions, and the quality of life [56].

The results of the study showed that neither of the two therapeutic strategies is more beneficial in the improvement of AF-associated symptoms. In PIAF, amiodarone led to restoration of SR in 23% of patients, while the remaining majority underwent at least one direct current cardioversion and as it has been shown, in 56% of patients who were successfully cardioverted, that SR could be maintained on a low-dose amiodarone treatment over the observation period, although the percentage occurring from the observations was smaller than what was initially expected, which was 70% [56]. However, the administration of amiodarone was terminated in 25% of patients due to presumed side effects, whereas there was no report of an amiodarone-associated proarrhythmic effect in PIAF, as supported by previous observations [57]. Patients in the rhythm control arm of the study had a better exercise tolerance when compared with those in the rate control arm [56]. This finding could be supported by a potential improvement in hemodynamics after the restoration of SR as has already been supported [58]. Despite this improvement there was no association with a development in the QoL when both groups were compared, confirming data occurring from a QoL-change-assessing trial concerning various treatments of AF [59]. In the rate control arm of the study, the improvement in the AF-associated symptoms was similar among the randomized patients and the hospital admissions due to the fact that the AF-related outcomes were fewer. The majority of the patients were treated with digoxin and the addition of diltiazem resulted in a small but significant decrease in mean heart rate and further control over the ventricular rate. The use of catheter modification of atrioventricular node was performed only in a very small amount of patients (Table 1) [56].

### 4.7. Okçün et al. Study

The purpose of this study was to determine whether a restoration and maintenance of SR will improve survival and exercise tolerance among patients with nonischemic cardiomyopathy and AF [60]. The patients were randomized to either the rhythm or the rate control group, and the follow up period lasted 3 years. The composite of embolism, death, and exercise capacity were the endpoints of this study in both groups. Of the 154 patients included in the study, 80 were randomized to the rhythm and 74 to the rate control group. Ten patients who were not successfully cardioverted were further assigned to the rate control group. In the rhythm control group, patients without any detectable thrombus in transesophageal echocardiography underwent cardioversion and received amiodarone during the follow-up period. In the rate control group, digoxin and metoprolol were used. In both groups, extra medication was administered as needed: digoxin, diuretics, an aldosterone antagonist, an angiotensin-converting enzyme inhibitor (lisinopril), b-blocker (metoprolol), or an angiotensin-II receptor antagonist (losartan) provided that there are no contraindications. Anticoagulation therapy (warfarin, targeted INR: 2.0–3.0) was used in the rate control group during the study period and in the rhythm control group up to the first month after cardioversion [60]. 

According to the results of the study, patients with chronic AF and nonischemic left ventricular systolic dysfunction may present with improved rates of mortality and exercise capacity after SR is restored and maintained (rhythm control group: six deaths and six thromboembolic events, rate control group: 36 deaths and 9 thromboembolic events). All of the strokes were reported after the discontinuation of warfarin in the rhythm control group, suggesting that patients with HF should continue receiving anticoagulation even after the restoration of SR (Table 1) [60].

### 4.8. Strategies of Treatment of Atrial Fibrillation (STAF) Study

The STAF study compared the strategies of rhythm and rate control in patients with persistent AF regarding mortality, QoL, adverse events of the disease, and therapy side effects. From January 1997 to August 1999, 200 patients were recruited in an interim analysis, with a minimum follow-up of 12 months or a primary end point [61]. Patients in the rhythm control groups were treated with external or internal cardioversion, along with a prophylaxis of AF recurrence including Class I antiarrhythmic agents or sotalol, and in patients with coronary heart disease or an impaired left ventricular function, b-blocker and/or amiodarone was used. Patients in the rate control group were receiving b-blockers, digitalis, and calcium antagonists, while atrioventricularnode ablation/modification with or without pacemaker implantation were used as an alternative. Oral anticoagulation was used in both treatment strategies. Patients with congestive heart failure, NYHA functional Class II or greater and LVEF < 45% (but not <20%) were eligible for the study. The primary end point of the study was a composite of death, stroke or transient ischemic attack, systemic embolism, and cardiopulmonary resuscitation. Secondary end points included syncope, severe bleeding, QoL, echocardiographic parameters, resting heart rate, and maintenance of SR at follow-up [61]. 

There was no difference in the primary and secondary end points between the two treatment strategies, except for the hospitalizations for cardiovascular reasons. Hospitalizations, basically concerning repeated cardioversions and initiation of antiarrhythmic treatment, were more frequent in the rhythmcontrol group. Additionally, there was an equal improvement in the QoL in both treatment groups although it was still lower than that of the healthy control group. It was also reported that rhythm control therapy showed no superiority in the maintenance of SR in a long term basis (23% in three years) when compared with rate control, although the number of patients observed was small [61]. It was possible to proceed to a comparison between different antiarrhythmics in the maintenance of SR in the STAF study [61]. In conclusion, under the conditions and the statistical limitations of this study, it is suggested that there was no benefit in choosing rhythm over rate control in patients with a high risk of arrhythmia recurrence. It is not yet clear whether the results in the rhythm control group would be better if the maintenance of SR was achieved in a higher proportion of patients (Table 1) [61].

### 4.9. Catheter Ablation versus Standard Conventional Therapy in Patients with Left Ventricular Dysfunction and Atrial Fibrillation (CASTLE-AF) Trial

The CASTLE-AF trial compared catheter ablation and pharmacological therapy for patients with AF and HF [62]. Over 300 patients with AF and NYHA functional Class II, III, or IV HF and with a left ventricular ejection fraction (LVEF) ≤ 35% were enrolled in the trial, all of whom had an implantable cardioverter defibrillator (ICD) or a cardiac resynchronization therapy defibrillator (CRT-D) and were randomized to either a catheter ablation group or pharmacological therapy group. Zhao et al. studied the potential superiority of rhythm control using AADsvs.the rate control in the terms of improving mortality and hospitalization for worsening HF [63]. The population of this sub-analysis consisted of patients who did not undergo an ablation procedure but were instead treated pharmacologically. Among 210 patients treated pharmacologically, 60 patients were in the rhythm control group and 150 were in the rate control group. Patients in the rhythm control group were on antiarrhythmics for the maintenance of SR and they were randomized in groups based on the ADDs administered (Class Ia, Ic, or III only). Patients in the rate control group were taking atrioventricular blocking agents: b-blockers, calcium antagonists, or digoxin. The primary endpoint of the study was a composite of all-cause mortality and hospitalization for worsening HF outcomes. Major secondary endpoints were all-cause mortality, hospitalization related to HF, cardiovascular-disease-related hospitalization, as well as any detected ventricular and/or atrial tachyarrhythmia [63]. 

In this subpopulation of the CASTLE-AF study, it was reported that pharmacological rhythm control strategies using AADs in patients with advanced HF was not superior to a pharmacological rate control strategy in the primary outcome of the study, and additionally the prevalence of ventricular arrhythmias was comparable between different treatment strategies [63]. A proportion of 86.7% of patients were prescribed amiodarone to maintain SR. The prolonged and short-term use of AADs is associated with risks of adverse events, annulling their potential beneficial effects in maintaining SR. Sixty-threeb-blockers were prescribed in 97.9% of the patients in the rate control arm and this was shown to reduce mortality in patients with left ventricular systolic dysfunction, as well as being associatedassociated with fewer side effects [64]. AADs led to a significant improvement in AF burden which was not linked with mortality or hospitalization benefits in comparison with the rate control strategy (Table 1) [63].

### 4.10. Duke Cardiovascular Disease Database Trial

A trial which was conducted on data collected from the Duke Cardiovascular Disease Database targeted the determination of whether a rhythm or a rate control strategy for patients with AF and diastolic heart failure affected survival [65]. For this study, diastolic heart failure was determined as the presence of signs and/or symptoms of congestive HF in a patient with a preserved ejection fraction >50% [66]. From January 1995 through June 2005, 382 patients were included in the study according to the criteria defined. Patients were randomized into two groups based on the treatment followed: in the rhythm control group, Class I or III AADs were used, and in the rate control group, where b-blockers, calcium-channel blockers, and/or digoxin were used. The endpoint of this study was all-cause mortality [65].

This study indicated that there was no statistical difference on survival between the rate and rhythm control strategies. Although there are not enough data describing the optimal management of AF in diastolic HF, there was a tendency of restoring and maintaining SR in such patients. It is assumed that SR benefits those patients because left ventricular filling in diastolic heart failure occurs primarily in late diastole and is dependent, unlike what happens in healthy hearts, on atrial contraction, which is not preserved in AF. Based on the data occurring from this study, there was no advantage of a rate over a rhythm control strategy for AF in patients with diastolic HF. Statistical adjustments though resulted in a better survival profile in patients with AF and HF with preserved ejection fraction, when it was managed with rhythm control (Table 1) [65].

### 4.11. Get with the Guidelines–Heart Failure (GWTG-HF) Analysis

A 2019 analysis on the Get with The Guidelines–Heart Failure (GWTG–HF) registry along with Medicare claims data from 2008 to 2014 described the treatments for rate versus rhythm control and the subsequent effects in patients with HFpEF and AF [67]. There were 15,682 patients ≥ 65 years of age in the GWTG–HF registry that were enrolled in the trial;1857 patients received rhythm control and 13,825 received rate control. In the rate control group, the used drugs were exclusively b-blockers, calcium channel blockers, and/or digoxin, and in the rhythm control group there were patients who had been treated with ablation/pulmonary vein isolation treatment, and they were in continuing therapy with amiodarone, sotalol, tikosyn, or other antiarrhythmic therapy or elective cardioversion or even with the previously mentioned rate control drugs. The primary outcome of the analysis was all-cause mortality. The secondary outcomes comprehended all-cause mortality or readmission, all-cause readmission, ischemic stroke, HF, other cardiovascular events, as well as bleeding readmissions [67].

In this population it was observed that rhythm control was associated with 1-year lower mortality when compared with the rate control treatment, even after risk adjustment. This result suggests that there might be an opportunity to improve outcomes in patients with HFpEF and AF, therefore future studies need to investigate this beneficial effect [67]. A number of studies have confirmed that rate control is not superior to rhythm control, and lenient rate control offers no benefit when compared to strict rate control in patients with AF; those results, however, cannot be generalized in patients with HFpEF [30,41,68]. Lam et al. [69] demonstrated the relevance of greater exercise intolerance, natriuretic peptide increase, and left atrial remodeling in patients with HFpEF and AF in comparison with those without AF, suggesting that the efficient control of AF may be beneficial in patients with HFpEF. Amiodarone and dofetilide, which were used in 67.2% and 2.0% of the patients respectively, are not contraindicated in HF patients despite limitations in their use. Although the contraindications associated with the use of Class Ic agents in patients with HFrEF are also extended in HFpEF patients, dronedarone and Class Ic agents were used in 11.4% of the patients of the rhythm control group. Another recent study that compared rate versus rhythm control in postoperative cardiovascular surgery showed no differences in outcomes between the two strategies through the next 2 months of follow-up [70]. In the GWTG–AF, 13.6% of patients who were in the rhythm control group had undergone cardioversion, indicating that the above post-cardioversion strategy is frequent [67]. In conclusion the non-superiority of either strategy in primary and secondary outcomes, the observed safety of rhythm control with a short-term use, along with the 6.7% lower all-cause mortality suggests that a possible benefit from rhythm control may nevertheless exist and should be further investigated (Table 1) [67].

**Table 1 medicina-58-00743-t001:** A summary of the studies comparing rhythm vs. rate control strategy.

Study [Supporting Reference]	Rate Control Intervention	Rhythm Control Intervention	Result of the Study
RACE [29,31,33]	Digitalis b-blocker nondihydropyridine calcium channel blocker	sotalol Class Ic agents amiodarone	- higher mortality and morbidity, thromboembolic complications, major fatal and nonfatal bleedings in rate control - prevention of a decline in LVEF in routine rate control - SR was associated with excellent survival and no improvement in chronic HF and left ventricular function in rhythm control
AF-CHF [37]	b-blocker with digitalis	amiodarone sotalol dofetilide	- no superiority of a rhythm control strategy against a rate control in cardiovascular or all cause mortality, worsening HF or stroke - results not extended in HFpEF patients - higher hospitalization rates in the rhythm control group in the first year - rate control potentially ideal in this population due to decreased hospitalization rates
AFFIRM [38,39,40,41,42,43,44,45,46]	b-blockers calcium channel blockers digoxin (alone or in combination)	Class Ic agents (flecainide and propafenone) sotalol disopyramide amiodarone	- no strategy associated with a more beneficial profile - symptomatic HF more common in the rate control - AF associated with poorer NYHA class, worsening HF, greater requirement for ACE inhibitors and diuretics, except for the group who crossed over from rhythm to rate control - highest rate of symptomatic HF in the patients who changed strategy between rate and rhythm control more than once - SR maintenance not beneficial in the prevention of embolism - warfarin in the rhythm control strategy with an additional risk of stroke
CAFE II [47]	b-blockers digoxin	amiodarone or external cardioversion	- improved QoL and left ventricular function in the rhythm control arm, not extended in exercise capacity - greater improvement in 1-year SR maintenance - decreased natriuretic peptides in patients with restored SR - no results in the effect of the intervention in the mortality rates - improvement of the HF symptoms through cardioversion, safe strategy not improving, however, long-term morbidity or mortality
HOTCAFE [53]	b-blockers calcium channel blockers digoxin (alone or in combination)	cardioversion prior to the AAD therapy: propafenone disopyramide sotalol amiodarone Class I agents b-blockers in clinical indication	- good management of symptoms and exercise tolerance in both groups - better rate control and decreased mean heart rate, higher maximal workload and longer exercise tolerance, increased left ventricular fractional shortening and decreased dimensions of right and left atria in rhythm control group at the end of follow-up - satisfactory ventricular response, fewer hospitalizations and less new-onset arrhythmias in the rate control group
PIAF [55,56]	diltiazem	electrical cardioversion prior to amiodarone	- neither of the two therapeutic strategies is more beneficial in the improvement of symtoms - better exercise tolerance in the rhytm control arm, with no association with QoL - the majority of patients were treated with digoxin and the addition of diltiazem resulted in a small but significant decrease in mean heart rate and further control in the ventricular rate
Okçün et al. [60]	digoxin metoprolol	cardioversion amiodarone in the follow-up	- improved rates of mortality and exercise capacity after SR is restored and maintained - patients with HF should continue receiving anticoagulation even after the restoration of SR
STAF [61]	b-blockers digitalis calcium channel blockers	Class I agents sotalol b-blocker and/or amiodarone in clinical indication	- no difference in the primary and secondary end points between the two treatment strategies, except for the hospitalizations for cardiovascular reasons - hospitalizations, basically concerning repeated cardioversions and initiation of antiarrhythmic treatment, were more frequent in the rhythmcontrol group - equal improvement in the QoL - no superiority of either strategy in the SR maintenance
CASTLE-AF [62,63]	b-blockers digitalis calcium channel blockers	Class Ia, Ic, III agents	- AAD rhythm control was not superior to a pharmacological rate control strategy in the primary outcome of the study and in the prevalence of ventricular arrhythmias - b-blockers were shown to reduce mortality in patients with left ventricular systolic dysfunction and are associated with fewer side effects - AADs led to a significant improvement in AF burden, not associated with mortality or hospitalization benefits in comparison with the rate control strategy
DUKE trial [65]	b-blockers digitalis calcium channel blockers	Class I or III agents	- no statistical difference on survival between the rate and rhythm control strategies - after statistical adjustments, a better survival profile in patients with AF and HFpEF was observed in the rhythm control group
GWTG–HF [67]	b-blockers digitalis calcium channel blockers	amiodarone sotalol tikosyn other AAD cardioversion prementioned rate control drugs	- rhythm control was associated with 1-year lower mortality compared with the rate control treatment, even after risk adjustment - safety of rhythm control in a short term use is observed along with the 6.7% lower all-cause mortality

Abbrevations: AAD (Anti-arrhythmic Drugs), AF (Atrial Fibrillation), ACE (Angiotensin Converting Enzyme), AF-CHF (Atrial Fibrillation and Congestive Heart Failure), AFFIRM (Atrial Fibrillation Follow-Up Investigation of Rhythm Management), CAFÉ II (Controlled study of rate versus rhythm control in patients with chronic atrial fibrillation and heart failure), CASTLE-AF (Catheter Ablation for AF with HF), GWTG-HF (Get With the Guidelines-Heart Failure), HF(Heart Failure), HFpEF(Heart Failure with preserved Ejection Fraction), HOT-CAFÉ (How to Treat Chronic Atrial Fibrillation Study), LVEF (Left ventricular ejection fraction), NYHA (New York Heart Association), QoL (Quality of Life), PIAF (Pharmacological Intervention in Atrial Fibrillation trial), RACE (RAte Control versus Electrical cardioversion), SR (Sinus Rhythm), STAF (Strategies of Treatment of Atrial Fibrillation).

## 5. Pharmacological Rhythm Control Therapy

The decision for the administration of a rhythm control treatment is often based on the objectives of controlling symptoms and reducing the need for hospitalization. AF is effectively controlled if recurrences are infrequent, well tolerated, self-terminating, and do not require hospitalization. An unsuccessful treatment is associated with the frequent development of symptoms, poor tolerance of the recurrent episodes of AF, and AF-related hospitalizations [8]. According to the Japanese Circulation Society (JCS), various types of Class I AADs are suggested for the pharmacological rhythm control treatment of paroxysmal lone AF, while Class III/IV AADs such as amiodarone, sotalol, and bepridil with/without aprindine are indicated for the treatment of persistent lone AF [71]. In concomitant AF with heart disease (hypertrophic, ischemic and/or HF), the JCS guidelines recommend class III/IV AADs for pharmacological rhythm control treatment, because Class I AADs have limited beneficial effects and a possibility of the development of proarrhythmic effects in AF coexisting with HF [71]. The guidelines of the American College of Cardiology/American Heart Association (ACC/AHA) for congestive heart failure support the withdrawal of the drugs associated with adverse HF outcomes [72].

### 5.1. Amiodarone

Amiodarone is the most commonly used AAD in the treatment of congestive HF, cardiac dysfunction and the first choice among AADs for the treatment of the combined AF and HF. The safety of amiodarone in HF has already been demonstrated [34,73], as well as its efficiency as an ADD for maintaining sinus rhythm [36,74,75]. While the restoration of SR seems to be independent of left ventricular function [76], a combined AFFIRM and AF-CHF analysis suggest that the efficacy of amiodarone in maintaining SR is also independent of LVEF, with no higher rates of recurrence being observed in the setting of HF [77]. Despite these observations mentioned above, amiodarone is not the most potent drug for acute pharmacological conversion. Amiodarone’s maximum efficiency is reached in up to 24 h [75] and its multiorgan adverse effects limit long-term therapy to the 15% of the patients receiving amiodarone [36]. The Canadian Trial of Atrial Fibrillation (CTAF) study included 403 patients with persistent AF, comparing and contrasting the treatment effects of amiodarone (201 patients), sotalol (101 patients) and propafenone (101 patients). According to the results the recurrence of AF was less frequent in the amiodarone group than in the sotalol and propafenone groups, but only 65% of patients in the amiodarone group presented without a recurrence of AF during the follow-up period of 16 months [36]. In other trials, where the rhythm control strategy was basically achieved using amiodarone, the number of patients in SR at the end of the study varied from 38 to 63.5% [30,56]. Large clinical trials (AFFIRM, STAF, PIAF, RACE) have shown that ventricular rate control is comparably as effective as rhythm control in terms of survival, QoL, anticoagulation, and embolism [78]. Despite the hypothesis that the maintenance of SR through AADs is preferred in patients with congestive HF due to a possible improvement of cardiac dysfunction, the results of the AF-CHF trial demonstrated that there was no difference in the survival rate between the two different treatment groups (rate and rhythm control strategy) at the end of the follow up period, revealing that even in patients with congestive HF, pharmacological rhythm control was equivalent to ventricular rate control [37]. The Stroke Prevention in Atrial Fibrillation (SPAF) study by Flaker et al., was applied on 1330 patients with AF and congestive HF, who were treated with Class Ia and Ic antiarrhythmics or amiodarone for the maintenance of SR [25]. The study reported an increased cardiac mortality among AF patients receiving antiarrhythmics (5%) when compared with those receiving a placebo (2.2%). It has also been shown that HF patients receiving antiarrhythmic therapy were at even greater relative risk of cardiac mortality, indicating that the risks of AAD might outweigh the potential benefit of the restoration of SR [25,79,80]. In the Congestive Heart Failure Survival Trial of Antiarrhythmic Therapy (CHF-STAT) trial, Deedwania et al., described the long-term effects of the administration of amiodarone on the morbidity and mortality in patients with congestive HF and AF in a 4-year period [35]. Of the 667 patients enrolled with congestive HF, 103 (15%) had AF, 51 were randomized to amiodarone and 52 to placebo. The analysis of total mortality during follow-up was associated with a significantly lower mortality rate in AF patients at baseline who converted to SR on amiodarone than those in whom SR was not restored. According to the data occurring from the study, patients with AF and HF were reported to spontaneously convert to SR more often during chronic amiodarone therapy. Additionally, those in SR at baseline are less likely to develop new-onset AF, and if AF occurs during the chronic amiodarone administration, the ventricular response remains significantly slower when compared to thepatientsreceiving placebo. The results of the trial led to the suggestion that, in patients with congestive HF, amiodarone could potentially be added as a first-line pharmacological treatment of AF [35].

### 5.2. Dronedarone

Recent antiarrhythmic drugs may be more beneficial and have less adverse effects in AF treatment of CHF patients [78]. The Permanent Atrial Fibrillation Outcome Study Using Dronedarone on Top of Standard Therapy (PALLAS) study was designed to evaluate the effects of dronedarone in the reduction of major vascular events or incident hospitalization for cardiovascular outcomes in patients with permanent AF [81]. Two-thirds of the patients had a history of HF. The endpoints of the study were a composite of stroke, MI, systemic embolism, hospitalization, and death from cardiovascular causes. The trial was terminated early due to safety issues. The reported results demonstrated that dronedarone increased the rates of stroke, heart failure, and death from cardiovascular causes in the study population, describing an interaction with digoxin resulting in increased levels of activity, as well as an association with arrhythmic events [81]. A placebo-controlled, double-blind, parallel arm trial to assess the efficacy of 400 mg dronedarone, called the Prevention of Cardiovascular Hospitalization or Death from Any Cause in Patients with Atrial Fibrillation/Atrial Flutter (ATHENA) trial, studied the role of dronedarone in the reduction of the composite outcome of hospitalization due to cardiovascular events or death in patients with AF [82]. A number of 4623 patients with paroxysmal or persistent AF were included in the trial, 21% of whom had a history of congestive HF (NYHA functional Class II–III) and 12% of whom had cardiac dysfunction (LVEF < 45%). Patients were randomized, in a 1:1 ratio, in two treatment groups to receive either dronedarone or placebo. Primary outcomes were all-cause mortality and hospitalization and cardiovascular events. Secondary study outcomes were all-cause mortality, death from cardiovascular causes, and first hospital admission due to cardiovascular events. In this study, dronedarone (800 mg/day) presented with a reduction in the primary endpoints when compared with the placebo group, and the beneficial effects of the drug were not limited to only those who converted to SR [82]. On the other hand, the Antiarrhythmic Trial with Dronedarone in Moderate to Severe CHF Evaluating Morbidity Decrease (ANDROMEDA) study included 627 patients with more severe congestive HF (NYHA functional Class III–IV) and cardiac dysfunction (LVEF < 35%) [83]. The primary outcome of the study was a composite of all-cause mortality and hospitalization for adverse HF events. Although the same dose of dronedarone (800 mg/day) as used in the ATHENA study was also prescribed in the ANDROMEDA study, in the population of the study, the treatment with dronedarone was associated with increased rates of mortality due to the worsening of cardiac function. There were also data indicating a decrease in the estimated glomerular filtration rate (GFR), with no clear evidence however of a deterioration of renal function [83]. Therefore, although dronedarone could be prescribed in patients with moderate cardiac dysfunction, it is not supposed to be able to be a substitute for amiodarone, especially in severe HF patients with reduced left ventricular systolic function. Moreover, an important matter is the interference with the renal function, the renal clearance of other drugs, as well as a potential threshold of GFR setting a contraindication for the administration of the drug.

### 5.3. Dofetilide

Dofetilide is an alternative AAD for rhythm control in patients with HF. Its safety has been demonstrated in the Danish Investigations of Arrhythmia and Mortality ON Dofetilide (DIAMOND) trial, designed to demonstrate the effect of dofetilide on the mortality in patients with HF and left ventricular systolic dysfunction with or without AF [84]. In a subpopulation of the DIAMOND trial, out of the total 3028 patients with severe congestive HF or recent MI, 506 (17%) had AF—atrial flutter [85]. A number of249 patients were randomized into the dofetilide treatment group and 257 into the placebo group. The results demonstrated that dofetilide is superior to placebo for the restoration and maintenance of SR in this subpopulation, as it was associated with lower morbidity and mortality rates as well as reduced hospitalization in patients where the SR was restored [85]. However, overall mortality was not reduced in the cohort or in the subgroup of patients with AF [84,85]. Dofetilide may serve as a potential alternative to amiodarone in the treatment of AF in patients with left ventricular dysfunction, despite the fact that QT prolongation and a subsequent risk of torsades de pointes was reported in 3% of HF patients in a 75% percentage within the first 3 days [84]. The initiation of treatment should be performed in a hospital, adjusted to the creatinine clearance, and with the QT interval being carefully monitored [8].

### 5.4. Sotalol

The double-blind Sotalol Amiodarone Atrial Fibrillation Efficacy Trial (SAFE-T) compared sotalol and amiodarone in the terms of the restoration and maintenance of SR in patients with AF and mostly normal LVEF [74]. A total of 6582 patients were enrolled in the study, 665 of whom underwent randomization: 267 to amiodarone, 261 to sotalol, and 137 to the placebo treatment group. The primary end point of the study was the recurrence of AF after the restoration of SR, and further endpoints were the changes in QoL and exercise ability. According to the results of the study, amiodarone and sotalol are equally efficacious in restoring SR in symptomatic and asymptomatic patients, and are associated with an improvement in QoL and exercise performance. Amiodarone is superior for maintaining sinus rhythm, but both drugs are equally efficient in patients with ischemic heart disease [74]. The d-sotalol isomer is appeared to increase mortality in patients with ischemic cardiomyopathy and reduced LVEF, as it has been reported in the SWORD trial [86]. This isomer lacks the b-blocker effect and retains the potassium-channel-blocking properties. The racemic d,l-sotalol might be beneficial in patients with HF and normal LVEF and in some patients with low LVEF and an ICD. A comparison between d,l-sotalol vs. placebo in patients with an ICD for secondary prevention showed that it was superior to placebo in ICD tolerance and that it reduced appropriate and inappropriate shocks with no effects on mortality, even in patients with LVEF < 30% [86]. While in the SWORD trial it was seen that the addition of another b-blocker to sotalol did not affect the combined outcome of mortality or ICD shock, it remains to be detected whether there is actually an additional benefit in the already-improved survival promoted by the b-blocker [8]. Sotalol should be administered carefully in the presence of cardiac disease [78], closely monitoring the renal function and the GFR, as a sotalol accumulation may lead to an increased risk of torsades [8].

The Rate Versus Catheter Ablation Rhythm Control in Patients with Heart Failure and High Burden Atrial Fibrillation (RAFT-AF) trial currently aims to determine whether the treatment of AF by catheter ablation, with or without antiarrhythmic drugs reduces all-cause mortality and hospitalizations for HF, in comparison with a rate control strategy in patients with HF and AF. A sample of 600 patients with NYHA functional Class II–III HF (HFrEF< 35%) or HFpEF) and a high burden of AF are included in the trial, randomized to either rate control or rhythm control in a 1:1 ratio [87]. The therapy in the rate control group includes optimal HF therapy and rate control strategies, while the rhythm control group therapy includes optimal HF therapy and one or more aggressive catheter ablation, with or without an additional AAD treatment. The primary outcome is a composite of all-cause mortality and hospitalization for HF events [87]. The suggestion that a more aggressive rhythm control strategy may prove to be superior to a rate control treatment is not supported by the current evidence, however. 

Rhythm control may be preferred as a treatment strategy in selected patients based on their clinical profile. For instance, patients with new-onset AF and new-onset systolic HF may be significantly benefited by a rhythm-control strategy and, by the maintenance of SR, the underlying cardiomyopathy could be reversed [8]. In patients with a newly diagnosed AF with acute decompensation of previously stable chronic heart failure, the maintenance of SR may be preferred if the ventricular rate is not excessively elevated [8]. A sub-analysis of AF-CHF suggested that patients with AF and HF presented with anxiety sensitivity, defined as the fear of sensations that occur in anxiety-provoking situations, such as palpitations or rapid heart rates, which may be benefited by a treatment strategy that is also guided by their personality traits. More specifically, in this sub-analysis, patients with a high anxiety sensitivity level that were randomized in a rhythm control therapy were presented with a reduction in cardiovascular mortality. No differences were observed between the two treatment strategies in patients with lower anxiety sensitivity levels [88]. Additionally, rhythm control may be optimal in patients with rapid AF despite the administration of a maximal tolerated dose of b-blockers, or in patients with paroxysmal rapid AF and episodes of sinus bradycardia [8].

In patients with chronic HF, a rhythm control strategy, either pharmacological or electrical, has not been proven to be superior to a rate control strategy in reducing mortality or morbidity [37]. In clinical practice, the risk for thromboembolic events should primarily be assessed unless hemodynamic instability requires immediate cardioversion. A low stroke risk is defined as AF with a known duration of <48 h, the absence of a mechanical valve, rheumatic heart disease, and recent stroke or transient ischemic attack. A high thromboembolic risk is associated with AF of unknown duration or a duration >48 h or any the above-mentioned high-risk features, and it requires either sufficient therapeutic anticoagulation for at least 3 weeks or no thrombus on transesophageal echocardiography prior to cardioversion [8]. Additionally, AADs should not be administered in patients with thromboembolic contraindications to cardioversion. According to the 2016 ESC guidelines, a rhythm control strategy is preferably applied on patients with a reversible secondary cause of AF (e.g., hyperthyroidism) or an obvious impulsive factor (e.g., recent pneumonia), as well as in patients with symptomatic AF after receiving optimal rate control and HF therapy [89]. Class I antiarrhythmic agents and dronedarone are associated with increased morbidity and mortality in patients with HF and AF,and therefore their use should be avoided [81,83,90]. Dofetilide and sotalol are indicated in special circumstances such as amiodarone intolerance or failure [79]. Amiodarone can restore SR in chronic AF patients, may reduce symptomatic paroxysmal AF, and contributes to the maintenance of SR after a spontaneous or electrical cardioversion [35,47,91,92]. The long-term administration of amiodarone should be consistently reassessed and justified [89]. A monitoring for bradyarrhythmias, thyroid and liver complications, and lung toxicity is also advised. The initiation of amiodarone in a patient on warfarin treatment requires the close monitoring of INR and an assessment of the optimal warfarin dosage [8].

If a patient presents with recurrent symptomatic AF despite the AAD treatment, a transition to rate control is optimal. If the rate control is ineffective in the alleviation of the AF symptoms, then repeated cardioversions with AAD dose adjustments may be attempted (amiodarone should be preferred if it is not already included in the initial treatment). If no one of the applied treatments is effective, AF ablation is still a potential option [8].

## 6. Pharmacological Rate Control Therapy

There is clinical evidence that slower heart rates in SR are associated with long-term survival. The Systolic Heart failure treatment with the If inhibitor Ivabradine (SHIFT) Trial was designed to define the effect of ivabradine in regards to guideline-based treatment on cardiovascular outcomes, symptoms, and QoL in patients with CHF and systolic dysfunction [93]. A number of 6558 patients were enrolled in the study and the analyzed population included 3268 in the ivabradine group and 3290 in the placebo group. The primary endpoint was the composite of cardiovascular mortality of hospitalization for adverse HF events. The results showed that ivabradine led to a significant reduction in the primary endpoints when added to guideline- and evidence-based treatment [93]. The effect of ivabradine in the reduction of the heart rate within 28 days of treatment and the consequent improvement in cardiovascular outcomes confirmed that a high heart rate is associated with an increased risk of HF, indicating that heart rate is an important target in HF treatment [94]. However, the negative association between fast heart rate pace in AF and cardiovascular outcomes has not yet been clarified [8]. The role of strict rate control and its possible beneficial effect in AF patients has been analyzed in the RACE II trial. A number of 68,614 patients with permanent AF were randomly assigned to either strict rate control (<80 beats per minute bpm at rest and <110 bpm with moderate exercise) or lenient rate control (<110 bpm at rest). The follow-up period was up to 3 years, and the reported results showed that lenient rate control was not inferior to strict rate control in the terms of cardiovascular death, HF hospitalization, systemic embolism, major bleeding, and arrhythmic events. These results should not be generalized in HF with impaired left ventricular systolic dysfunction. In a two-year observation of the PRIME II study population, patients with AF and advanced HF (NYHA functional Class III or IV) were in optimal treatment for CHF and the results of the study indicated that, at baseline, those with higher heart rates are comparable to those with low heart rates, whereas lower heart rates at baseline were associated with a poorer prognosis [95]. Another study by Silvet et al. described the effects of QoL and exercise tolerance in patients with chronic AF and HF after the administration of strict HR control [96]. In a population of 20 patients treated with increasing doses of metoprolol succinate to a targeted heart rate <70 bpm, aggressive HR control was not easily achieved due to patient intolerance of increasing doses of b-blockers. A moderate decrease in HR was not associated with improved outcomes [96]. A combined sub-study of the AFFIRM and AF-CHF trials assessed the association between heart rate at rest, in SR and in AF, and the subsequent cardiovascular outcomes, including all-cause mortality and cardiovascular hospitalizations, in patients with a history of paroxysmal or persistent AF [97]. An elevated baseline heart rate in SR was reported to be significantly associated with mortality. On the contrary, a rapid heart rate at baseline in AF was associated with incident hospitalizations but not mortality [97]. Additional evidence is required to determine the benefit of modified heart rates in the cardiovascular outcomes in this patient population. 

An evaluation of ventricular rate control from the radial pulse is not optimal, especially in HF patients, as the ventricular response does not always generate a palpable pulse. An electrocardiographic documentation of rate control is necessary; a wearable device used for routine monitoring, however, has not yet been established. The 2016 ESC HF guidelines proposed the optimal ventricular rate at rest in patients with AF and HF should be established between 60 and 100 bpm [30,38,98,99]. A resting ventricular rate of 110 bpm could also be optimal as a threshold target for the rate control therapy [68,100,101]. Τhe ventricular rate during light exercise is estimated to be <110 bpm [102]. Ventricular rates <70 bpm are linked with worse outcomes [102]. This may explain why the use of b-blockers in guideline-targeted doses failed to reduce morbidity or mortality in patients with HFrEF and AF [103], and may as well explain the adverse outcomes associated with digoxin, as have been described in some observational studies of AF [104,105,106].

B-blockers appear safe and efficient as a first-line rate control treatment, although it is not clear whether they reduce morbidity and mortality in patients with AF [89]. The variable pharmacodynamics of different b-blockers, carvedilol, bisoprolol, or metoprolol succinate, present with survival benefits and therefore should be preferred [107]. Carvedilol is reported to achieve fewer rate-slowing effects, especially if specific genetic polymorphisms exist [108,109,110]. A sub-analysis conducted by Joglar et al., on the US Carvedilol Heart Failure Trials data evaluated the effects of carvedilol in the hemodynamic condition and survival in patients with AF and symptomatic congestive HF [111]. The results demonstrated that carvedilol improved LVEF and there was a trend towards the reduction of mortality and hospitalization in the study population [111]. Fung et al. evaluated the role of beta-blockade in a similar clinical benefit for HF and AF patients as in those with HF in SR. The patients evaluated received metoprolol or carvedilol at titrated doses and the primary end points included an assessment of symptoms, exercise capacity and LVEF. Metoprolol and carvedilol administration was reported to improve LVEF in CHF patients in both AF and SR conditions, whereas the improvement in exercise capacity and QoL was only observed in patients in SR [112]. Stankovic et al., analyzed data from The Cardiac Insufficiency Bisoprolol Study in Elderly (CIBIS-ELD) to compare the response to titrated doses of carvedilol or bisoprolol in elderly CHF patients with AF vs. those in SR. This study is the first head-to-head comparison of in patients with chronic HF and coexisting AF. Elderly patients with chronic HF and AF presented with comparable clinical benefits from b-blocker titration as those in SR, in terms of LVEF improvement, exercise capacity, NYHA class, and QoL. Patients with AF were successfully treated with higher beta-blocker doses than those in SR, which appears to be associated with a higher baseline heart rate. Additionally, patients with a higher baseline heart rate presented with larger reductions in heart frequency, regardless of rhythm [113]. In contrast, a sub-analysis of the Cardiac Insufficiency Bisoprolol Study II (CIBIS II) trial demonstrated the similar benefit of bisoprolol when compared with placebo in promoting both the baseline heart rate and heart rate changes over time, as well as no significant benefit of bisoprolol being observed in terms of mortality, cardiovascular mortality, and hospitalization for HF in patients with AF [114]. In a sub-analysis of the Study of Effects of Nebivolol Intervention on Outcomes and Rehospitalization in Seniors with Heart Failure (SENIORS) trial, the administration of nebivolol in elderly patients with HF and AF was evaluated in the terms of cardiovascular outcomes. In patients with AF, nebivolol was less beneficial in all-cause mortality and cardiovascular hospitalizations when compared with patients in SR, irrespective of ejection fraction [115]. Additionally, the outcomes were comparable in AF patients with either HFrEF or HFpEF [115]. A sub-analysis of the Metoprolol CR/XL Randomized Intervention Trial in Chronic Heart Failure (MERIT-HF) did not reveal an interaction between metoprolol and mortality in patients with AF and HF, while the use of metoprolol in patients in SR at baseline reduced the incidence of AF in the follow-up [116].

Digitalis has been long used in the treatment of patients with HF. The Digitalis Investigation Group (DIG) trial studied the effect of digoxin on mortality and hospitalizations in patients with HF. The trial demonstrated reduced hospitalizations in patients with reduced LVEF and SR but no reduction in mortality was confirmed [117]. When administrated to AF patients, ventricular rate slowing is caused by a parasympathetic effect [118]. A multivariate analysis of the AFFIRM trial, after correcting for clinical characteristics and comorbidities, regardless of gender and the presence or absence of HF, revealed a significant increase in all-cause mortality in AF patients [119]. These findings discourage the use of digoxin in patients with AF and HF, and possibly it could be used as a second line agent when the use of b-blockers is not optimal or the number of hospitalizations due to HF is required to be decreased [8].

B-blockers combined with digoxin may also be used in the control of the ventricular rate [48]. Khand et al. [48] compared the effects of digoxin, carvedilol, alone and in combination in patients with HF and persistent AF. Forty-seven patients were randomized into two groups: 24 into the carvedilol group and 23 into the placebo group. The primary endpoints of the trial were left ventricular function, ventricular rate control, symptoms, and exercise capacity. The combination of digoxin and carvedilol was proven to reduce symptoms, improve ventricular function, and improve ventricular rate control, showing better results than either agent alone. Few differences were reported between the use of carvedilol and digoxin as single agents. This study also indicated the continued value of digoxin along with a b-blocker for the treatment of persistent AF in the setting of HF [48]. Beta-blockers reduce the ventricular rate during exercise, while the effects of digoxin are greater at night [48]. The persistence of high ventricular rates may be associated with thyrotoxicosis or excessive sympathetic activity due to congestion that might respond to diuresis [89].

Amiodarone has been reported to be included in the rate control therapy; in acute decompensated HF it is better tolerated than b-blockers as an agent and is more effective than digitalis. Amiodarone would be more helpful in patients with new-onset AF <48 h, while the embolism risk assessment is the same as previously mentioned in the rhythm control strategy. According to the ANDROMEDA and PALLAS trials, dronedarone should also be avoided in patients with HF and permanent AF [81,83]. Despite the reduction that amiodarone and nondihydropyridine CCBs can cause in the ventricular rate, they have more adverse effects due to their negative inotropic effects and it is suggested that they should be avoided in patients with HFrEF and possibly also in patients with HFpEF and HFmrEF [89].

Patients treated with a non-strict rate-control strategy that remain symptomatic, with a lower heart rate at rest, could be pursued, and with persistent symptoms at an optimal resting heart rate, rate control during exertion may also be optimized. Alternatively, a rhythm control therapy may be considered, if it has not already been attempted [8]. Frequently the ventricular rate cannot be reduced below 100–110 bpm using pharmacological treatment alone, therefore atrioventricular node ablation with ventricular pacing may be consideredin these cases [89].

## 7. Conclusions

A great effort has been directed to the improvements in the management of AF and HF as separate disease entities, and despite the gradual progress in the determination of the pathophysiology and the interrelated mechanisms of their coexistence, this has provided the clinical practice only with a few randomized trials and data concerning the optimal therapeutic strategy in the concomitant of AF and HF. Neither the rhythm nor rate control strategy has been proven so far to be superior in the management of such patients, although many trials adjusting their analyses according to the characteristics of the patients have demonstrated a more beneficial role of either of the two strategies in certain endpoints. Therefore, the ideal therapeutic strategy is applied based on the approved guidelines and the results of large, randomized trials, along with the clinical experience and the medical judgment of the treating team. Further trials and studies are required to clarify the interaction between the rate and rhythm control strategy in AF and HF and the optimal combination of the existing therapies in order to achieve the greatest improvement in cardiovascular outcomes, hospitalization, QoL, and survival.

## Data Availability

Not applicable.

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
