# Peer review of "Pharmacologic Rate versus Rhythm Control for Atrial Fibrillation in Heart Failure Patients"

_medicina, 2022, doi:10.3390/medicina58060743_

Round 1

Reviewer 1 Report

I have read this article with great pleasure.   The article is quite large in volume and requires some time to read it. That doesn't mean it's a remark.   The authors have fully disclosed the current state of the issue on this topic.   I have no fundamental comments to the article. Although I would recommend to check the accuracy of the links in the table. For example, the RACE study, references [29-36]. Do all these works relate to this study? Since reference 30 (which is below in the table) is mentioned in the GWTG-HF study [30,41,67-70].

Reviewer 2 Report

The papaer is an interesting review on the Pharmacologic rate versus rhythm control for atrial fibrillation in heart failure patients, I think it can be published with small revisions, a summary of the often too extensive contents would be useful 

Reviewer 3 Report

In this review, authors discuss the pathophysiology and therapeutics of atrial fibrillation in the context of heart failure.  

The introduction (section1) needs some work. Since there are recent reviews in the literature about this topic, for instance reference 2 (Carlisle et al. JACC: Heart Failure 2019), authors should mention why their review is interesting by highlighting the strengths. This information is relevant (and is missing) also in the abstract. In addition, in the introduction an overview of the topics to be discussed would be appreciated.

The layout of Table 1 can be improved by using lines or two different colors to separate the studies.

In Table 1, fist result of the RACE study, the comment “equal benefit in mortality and morbidity” does not correspond to the comparison between rate control and rhythm control groups and is misleading.

Section 4 would benefit from a short paragraph (before subsection 4.1) putting into context the studies discussed by the authors.
